# MIRA: Mental Imagery for Robotic Affordances

**Lin Yen-Chen[1], Pete Florence[2], Andy Zeng[2], Jonathan T. Barron[2],**
**Yilun Du[1], Wei-Chiu Ma[1], Anthony Simeonov[1], Alberto Rodriguez Garcia[1], Phillip Isola[1]**

[1]MIT          [2]Google

**Abstract:** Humans form mental images of 3D scenes to support counterfactual imagination, planning, and motor control. Our abilities to predict the appearance and affordance of the scene from previously unobserved viewpoints aid us in performing manipulation tasks (e.g., 6-DoF kitting) with a level of ease that is currently out of reach for existing robot learning frameworks. In this work, we aim to build artificial systems that can analogously plan actions on top of imagined images. To this end, we introduce Mental Imagery for Robotic Affordances (MIRA), an action reasoning framework that optimizes actions with novel-view synthesis and affordance prediction in the loop. Given a set of 2D RGB images, MIRA builds a consistent 3D scene representation, through which we synthesize novel orthographic views amenable to pixel-wise affordances prediction for action optimization. We illustrate how this optimization process enables us to generalize to unseen out-of-plane rotations for 6-DoF robotic manipulation tasks given a limited number of demonstrations, paving the way toward machines that autonomously learn to understand the world around them for planning actions.

**Keywords:** Neural Radiance Fields, Rearrangement, Robotic Manipulation

## 1 Introduction

Suppose you are shown a small, unfamiliar object and asked if it could fit through an M-shaped slot. How might you solve this task? One approach would be to "rotate" the object in your mind's eye and see if, from some particular angle, the object's profile fits into an M. To put the object through the slot would then just require orienting it to that particular imagined angle. In their famous experiments on "mental rotation", Shepard & Metzler argued that this is the approach humans use when reasoning about the relative poses of novel shapes [1]. Decades of work in psychology have documented numerous other ways that "mental images", i.e. pictures in our heads, can aid human cognition [2]. In this paper, we ask: can we give robots a similar ability, where they use mental imagery to aid their spatial reasoning?

Fortunately, the generic ability to perform imagined translations and rotations of a scene, also known as *novel view synthesis*, has seen a recent explosion of research in the computer vision and graphics community [3, 4, 5]. Our work builds in particular upon Neural Radiance Fields (NeRFs) [6], which can render what a scene would look like from any camera pose. We treat a NeRF as a robot's "mind's eye", a virtual camera it may use to imagine how the scene would look were the robot to reposition itself. We couple this ability with an affordance model [7], which predicts, from any given view of the scene, what actions are currently afforded. Then the robot must just search, in its imagination, for the mental image that best affords the action it wishes to execute, then execute the action corresponding to that mental image. We test this framework on 6-DoF rearrangement tasks [8], where the affordance model simply predicts, for each pixel in a given camera view, what is the action-value of picking (or placing) at that pixel's coordinates. Using NeRF as a virtual camera for this task has several advantages over prior works which used physical cameras:

- **Out-of-plane rotation.** Prior works have applied affordance maps to 2-dimensional top-down camera views, allowing only the selection of top-down picking and placing actions [7, 9, 10]. We instead formulate the pick-and-place problem as an action optimization process that searches across different novel synthesized views and their affordances of the scene. We demonstrate that this optimization process can handle the multi-modality of picking and placing while naturally supporting actions that involve out-of-plane rotations.

6th Conference on Robot Learning (CoRL 2022), Auckland, New Zealand.

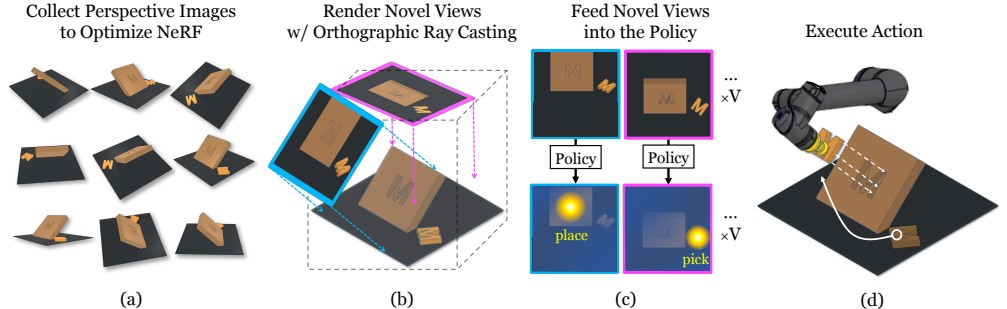

| (a) | (b) | (c) | (d) |
|-----|-----|-----|-----|
| Collect Perspective Images to Optimize NeRF | Render Novel Views w/ Orthographic Ray Casting | Feed Novel Views into the Policy | Execute Action |

**Figure 1: Overview of MIRA.** (a) Given a set of multi-view RGB images as input, we optimize a neural radiance field representation of the scene via volume rendering with perspective ray casting. (b) After the NeRF is optimized, we perform volume rendering with orthographic ray casting to render the scene from $V$ viewpoints. (c) The rendered orthographic images are fed into the policy for predicting pixel-wise action-values that correlate with picking and placing success. (d) The pixel with the highest action-value is selected, and its estimated depth and associated view orientation are used to parameterize the robot's motion primitive.

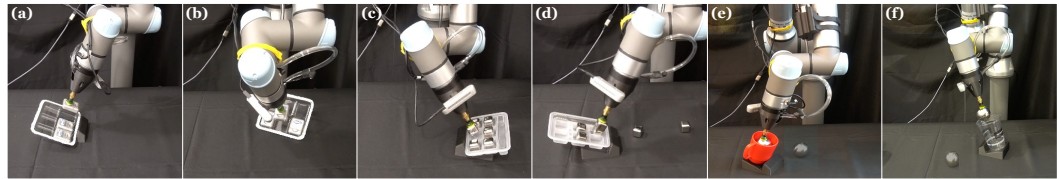

**Figure 2: Real-world Results.** MIRA is sample efficient in learning vision-based 6-DoF manipulation tasks: packing flosses (a-b), packing metal cubes (c-d), and putting the metal sphere into cups (e-f). These tasks are challenging for methods that rely on 3D sensors as objects contain thin structures or are composed of specular or semi-transparent material. MIRA is able to solve these tasks because it only requires RGB inputs.

- **Orthographic ray casting.** A NeRF trained with images from consumer cameras can be used to synthesize novel views from *novel kinds of cameras* that are more suitable to action reasoning. Most physical cameras use perspective projection, in which the apparent size of an object in the image plane is inversely proportional to that object's distance from the camera — a relationship that any vision algorithm must comprehend and disentangle. NeRF can instead create images under other rendering procedures; we show that orthographic ray casting is particularly useful, which corresponds to a non-physical "camera" that is infinitely large and infinitely distant from the scene. This yields images in which an object's size in the image plane is invariant to its distance from the camera, and its appearance is equivariant with respect to translation parallel to the image plane. In essence, this novel usage of NeRF allows us to generate "blueprints" for the scene that complement the inductive biases of algorithms that encode translational equivariance (such as ConvNets).

- **RGB-only.** Prior rearrangement methods [11, 12, 13] commonly require 3D sensors (e.g. via structured light, stereo, or time-of-flight), and these are error-prone when objects contain thin structures or are composed of specular or semi-transparent materials—a common occurrence (see Fig. 2 for examples). These limitations drastically restrict the set of tasks, objects, and surfaces these prior works can reason over.

We term our method Mental Imagery for Robotic Affordances, or MIRA. To test MIRA, we perform experiments in both simulation and the real world. For simulation, we extend the Ravens [9] benchmark to include tasks that require 6-DoF actions. Our model demonstrates superior performance to existing state-of-the-art methods for object rearrangement [14, 9], despite not requiring depth sensors. Importantly, the optimization process with novel view synthesis and affordance prediction in the loop enables our framework to generalize to out-of-distribution object configurations, where the baselines struggle. In summary, we contribute (i) a framework that uses NeRFs as the scene representation to perform novel view synthesis for precise object rearrangement, (ii) an orthographic ray casting procedure for NeRFs rendering that facilitates the policy's translation equivariance, (iii) an extended benchmark of 6-DoF manipulation tasks in Ravens [9], and (iv) empirical results on a broad range of manipulation tasks, validated with real-robot experiments.

## 2 Related Works

### 2.1 Vision-based Manipulation.

**Object-centric.** Classical methods in visual perception for robotic manipulation mainly focus on representing instances with 6-DoF poses [15, 16, 17, 18, 19, 20, 21]. However, 6-DoF poses cannot represent the states of deformable objects or granular media, and cannot capture large intra-category variations of unseen instances [11]. Alternative methods that represent objects with dense descriptors [22, 23, 24] or keypoints [11, 25, 26, 27] improve generalization, but they require a dedicated data collection procedure (e.g., configuring scenes with single objects).

**Action-centric.** Recent methods based on end-to-end learning directly predict actions given visual observations [28, 29, 30, 31, 32, 33]. These methods can potentially work with deformable objects or granular media, and do not require any object-specific data collection procedures. However, these methods are known to be sample inefficient and challenging to debug. Recently, several works [9, 13, 34, 35, 36, 37, 38] have proposed to incorporate spatial structure into action reasoning for improved performance and better sample efficiency. Among them, the closest work to ours is Song et al. [13] which relies on view synthesis to plan 6-DoF picking. Our work differs in that it 1) uses NeRF whereas [13] uses TSDF [39], 2) does not require depth sensors, 3) uses orthographic image representation, 4) does not directly use the camera pose as actions, and 5) shows results on rearrangement tasks that require both picking and placing.

### 2.2 Neural Fields for Robotics

Neural fields have emerged as a promising tool to represent 2D images [40], 3D geometry [41, 42], appearance [6, 43, 44], touch [45], and audio [46, 47]. They offer several advantages over classic representations (e.g., voxels, point clouds, and meshes) including reconstruction quality, and memory efficiency. Several works have explored the usage of neural fields for robotic applications including localization [48, 49], SLAM [50, 51, 52], navigation [53], dynamics modeling [54, 55, 56, 57], and reinforcement learning [58]. For robotic manipulation, GIGA [59] and NDF [12] use occupancy networks' feature fields to help action prediction. However, both of them rely on depth cameras to perceive the 3D geometry of the scene, while MIRA does not require depth sensors and thus can handle objects with reflective or thin-structured materials. Dex-NeRF [60] infers the geometry of transparent objects with NeRF and determines the grasp poses with Dex-Net [30]. However, it only predicts the 3-DoF grasping pose and does not provide a solution for pick-conditioned placing; NeRF-Supervision [61] uses NeRF as a dataset generator to learn dense object descriptors for picking but not placing. In contrast, MIRA is capable of predicting both 6-DoF picking and pick-conditioned placing. Model-based methods [54, 55] use NeRFs as decoders to learn latent state representations for model predictive control; NeRF-RL [58] instead uses the learned latent state representations for downstream reinforcement learning. Compared to these works, MIRA uses imitation learning to acquire the policy and thus avoid the conundrum of constructing reward functions. Furthermore, MIRA enjoys better sample efficiency due to the approximate 3D rotational equivariance via novel-view synthesis. The last but not least, MIRA demonstrates real-world results on three 6-DoF kitting tasks while these works primarily focus on simulation benchmark.

## 3 Method

Our goal is to predict actions $a_t$, given RGB-only visual observations $o_t$, and trained from only a limited number of demonstrations. We parameterize our action space with two-pose primitives $a_t = (\mathcal{T}_{\text{pick}}, \mathcal{T}_{\text{place}})$, which are able to flexibly parameterize rearrangement tasks [9]. This problem is challenging due to the high degrees of freedom of $a_t$ (12 degrees of freedom for two full SE(3) poses), a lack of information about the underlying object state (such as object poses), and limited data. Our method (illustrated in Fig. 1) factorizes action reasoning into two modules: 1) a continuous neural radiance field that can synthesize virtual views of the scene at novel viewpoints, and 2) an optimization procedure which optimizes actions by predicting per-pixel affordances across different synthesized virtual pixels. We discuss these two modules in Sec. 3.1 and Sec. 3.2 respectively, followed by training details in Sec. 3.3.

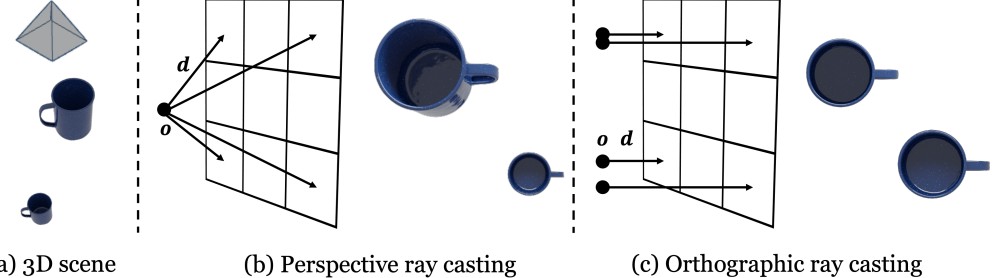

| (a) 3D scene | (b) Perspective ray casting | (c) Orthographic ray casting |

**Figure 3: Perspective vs. Orthographic Ray Casting.** (a) A 3D world showing two objects, with the camera located at the top. (b) The procedure of perspective ray casting and a perspective rendering of the scene. The nearby object is large, the distant object is small, and both objects appear "tilted" according to their position. (c) The procedure of orthographic ray casting and an orthographic rendering of the scene, which does not correspond to any real consumer camera, wherein the size and appearance of both objects are invariant to their distances and equivariant to their locations. By using NeRF to synthesize these orthographic images, which correspond to non-physical cameras, we are able to construct RGB inputs that are equivariant with translation.

## 3.1 Scene Representation with Neural Radiance Field

To provide high-fidelity novel-view synthesis of virtual cameras, we represent the scene with a neural radiance field (NeRF) [6]. For our purposes, a key feature of NeRF is that it *renders individual rays (pixels) rather than whole images*, which enables flexible parameterization of rendering at inference time, including camera models that are non-physical (e.g., orthographic cameras) and not provided in the training set.

To render a pixel, NeRF casts a ray $\mathbf{r}(t) = \mathbf{o} + t\mathbf{d}$ from some origin $\mathbf{o}$ along the direction $\mathbf{d}$ passing through that pixel on an image plane. In particular, these rays are casted into a field $F_\Theta$ whose input is a 3D location $\mathbf{x} = (x,y,z)$ and unit-norm viewing direction $\mathbf{d}$, and whose output is an emitted color $c = (r,g,b)$ and volume density $\sigma$. Along each ray, $K$ discrete points $\{\mathbf{x}_k = \mathbf{r}(t_k)\}_{k=1}^K$ are sampled for use as input to $F_\Theta$, which outputs a set of densities and colors $\{\sigma_k, \mathbf{c}_k\}_{k=1}^K = \{F_\Theta(\mathbf{x}_k, \mathbf{d})\}_{k=1}^K$. Volume rendering [62] with a numerical quadrature approximation [63] is performed using these values to produce the color $\hat{\mathbf{C}}(\mathbf{r})$ of that pixel:

$$\hat{\mathbf{C}}(\mathbf{r}) = \sum_{k=1}^K T_k \left(1 - \exp\left(-\sigma_k(t_{k+1} - t_k)\right)\right)\mathbf{c}_k, \quad T_k = \exp\left(-\sum_{k'<k} \sigma_{k'}(t_{k'+1} - t_{k'})\right). \tag{1}$$

where $T_k$ represents the probability that the ray successfully transmits to point $\mathbf{r}(t_k)$. At the beginning of each pick-and-place, our system takes multi-view posed RGB images as input and optimizes $\Theta$ by minimizing a photometric loss $\mathcal{L}_{\text{photo}} = \sum_{\mathbf{r} \in \mathcal{R}} \|\hat{\mathbf{C}}(\mathbf{r}) - \mathbf{C}(\mathbf{r})\|_2^2$, using some sampled set of rays $\mathbf{r} \in \mathcal{R}$, where $\mathbf{C}(\mathbf{r})$ is the observed RGB value of the pixel corresponding to ray $\mathbf{r}$ in an input image. In practice, we use instant-NGP [64] to accelerate NeRF training and inference.

**Orthographic Ray Casting.** In Fig. 3 we illustrate the difference between perspective and orthographic cameras. Though renderings from a NeRF $F_\Theta$ are highly realistic, the perspective ray casting procedure used by default in NeRF's volume rendering, which we visualize in Fig. 3(b), may cause scene content to appear distorted or scaled depending on the viewing angle and the camera's field of view — more distant objects will appear smaller in the image plane. Specifically, given a pixel coordinate $(u,v)$ and camera pose $(\mathbf{R}, \mathbf{t})$, NeRF forms a ray $\mathbf{r} = (\mathbf{o}, \mathbf{d})$ using the perspective camera model:

$$\mathbf{o} = \mathbf{t}, \qquad \mathbf{d} = \mathbf{R}\begin{bmatrix}(u - c_x)/f_x \\ (v - c_y)/f_y \\ 1\end{bmatrix}. \tag{2}$$

This model is a reasonable proxy for the geometry of most consumer RGB cameras, hence its use by NeRF during training and evaluation. However, the distortion and scaling effects caused by perspective ray casting degrades the performance of the downstream optimization procedure that takes as input the synthesized images rendered by NeRF, as we will demonstrate in our results (Sec. 3.4). To address this issue, we modify the rendering procedure of NeRF after it is optimized by replacing perspective ray casting

with orthographic ray casting:

$$\mathbf{o} = \mathbf{t} + \mathbf{R} \begin{bmatrix} (u-c_x)/f_x \\ (v-c_y)/f_y \\ 0 \end{bmatrix}, \qquad \mathbf{d} = \mathbf{R} \begin{bmatrix} 0 \\ 0 \\ 1 \end{bmatrix}. \tag{3}$$

To our knowledge, our work is the first to demonstrate that when a NeRF model is trained on perspective cameras, it may directly be utilized to render orthographic images. Such a result is both non-obvious and surprising because rays are now rendered using a different volume rendering procedure from the one used during training time, and the orthographic rendering process essentially corresponds to an out-of-distribution rendering test on the NeRF model.

We visualize this procedure in Fig. 3(c). Orthographic ray casting marches parallel rays into the scene, so that each rendered pixel represents a parallel window of 3D space. This property removes the dependence between an object's appearance and its distance to the camera: an object looks the same if it is either far or nearby. Further, as all rays for a given camera rotation $\mathbf{R}$ are parallel, this provides equivariance to the in-plane camera center $c_x, c_y$. These attributes facilitate downstream learning to be equivariant to objects' 3D locations and thereby encourages generalization. As open-source instant-NGP [64] did not support orthographic projection, we implement this ourselves as a CUDA kernel (see Supp.).

Our decision of choosing an orthographic view of the scene draws inspiration from previous works that have used single-view orthographic scene representations [16, 9], but critically differs in the following two aspects: (a) we create orthographic scene representations by casting rays into a radiance field rather than by point-cloud reprojection, thereby significantly reducing image artifacts (see Supp.); (b) we rely on multi-view RGB images to recover scene geometry, instead of depth sensors.

### 3.2 Policy Representation: Affordance Raycasts in a Radiance Field

To address *SE*(3)-parameterized actions, we formulate action selection as an optimization problem over synthesized novel-view pixels and their affordances. Using our NeRF-based scene representation, we densely sample $V$ camera poses around the workspace and render images $\hat{I}_{v_t} = F_{\Theta}(\mathcal{T}_{v_t})$ for each pose, $\forall v_t = 0, 1, \cdots, V$. One valid approach is to search for actions directly in the space of camera poses for the best $\mathcal{T}_{v_t}$, but orders of magnitude computation may be saved by instead considering actions that correspond to *each pixel* within each image, and sharing computation between all pixels in the image (e.g., by processing each image with just a single pass through a ConvNet). This (i) extends the paradigm of pixel-wise affordances [7] into full 6-DOF, novel-view-enabled action spaces, and (ii) alleviates the search over poses due to translational equivariance provided by orthographic rendering (Sec. 3.1).

Accordingly, we formulate each pixel in each synthesized view as parameterizing a robot action, and we learn a dense action-value function $E$ which outputs per-pixel action-values of shape $\mathbb{R}^{H \times W}$ given a novel-view image of shape $\mathbb{R}^{H \times W \times 3}$. Actions are selected by simultaneously searching across all pixels $\mathbf{u}$ in all synthesized views $v_t$:

$$\mathbf{u}_t^*, v_t^* = \underset{\mathbf{u}_t, v_t}{\arg\min} \, E(\hat{I}_{v_t}, \mathbf{u}_t), \quad \forall v_t = 0, 1, \cdots, V \tag{4}$$

where the pixel $\mathbf{u}_t^*$ and the associated estimated depth $d(\mathbf{u}_t^*)$ from NeRF are used to determine the 3D translation, and the orientation of $\mathcal{T}_{v_t^*}$ is used to determine the 3D rotation of the predicted action. Our approach employs a single ConvNet that is shared across all views and use multiple strategies for equivariance: 3D translational equivariance is in part enabled by orthographic raycasting and synergizes well with translationally-equivariant dense model architectures for $E$ such as ConvNets [65, 66], meanwhile 3D rotational equivariance is also encouraged, as synthesized rotated views can densely cover novel orientations of objects.

While the formulation above may be used to predict the picking pose $\mathcal{T}_{\text{pick}}$ and the placing pose $\mathcal{T}_{\text{place}}$ independently, intuitively the prediction of $\mathcal{T}_{\text{pick}}$ affects the prediction of $\mathcal{T}_{\text{place}}$ due to the latter's geometric dependence on the former. We therefore decompose the action-value function into (i) picking and (ii)

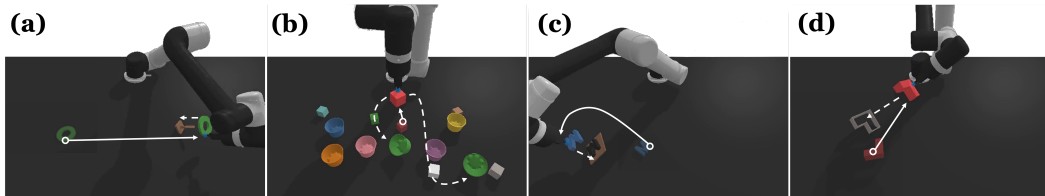

**Figure 4: Simulation qualitative results.** MIRA only requires RGB inputs and can solve different 6-DoF tasks: (a) hanging-disks, (b) place-red-in-green, (c) stacking-objects, and (d) block-insertion.

pick-conditioned placing, similar to prior work [9]:

$$\mathbf{u}^*_{\text{pick}}, v^*_{\text{pick}} = \underset{\mathbf{u}_{\text{pick}}, v_{\text{pick}}}{\arg\min} \, E_{\text{pick}}(\hat{I}_{v_{\text{pick}}}, \mathbf{u}_{\text{pick}}), \qquad\qquad \forall v_{\text{pick}} = 0, 1, \cdots, V \qquad (5)$$

$$\mathbf{u}^*_{\text{place}}, v^*_{\text{place}} = \underset{\mathbf{u}_{\text{place}}, v_{\text{place}}}{\arg\min} \, E_{\text{place}}(\hat{I}_{v_{\text{place}}}, \mathbf{u}_{\text{place}} | \mathbf{u}^*_{\text{pick}}, v^*_{\text{pick}}), \qquad\qquad \forall v_{\text{place}} = 0, 1, \cdots, V \qquad (6)$$

where $E_{\text{place}}$ uses the Transport operation from Zeng et al. [9] to convolve the feature map of $\hat{I}_{v^*_{\text{pick}}}$ around $\mathbf{u}^*_{\text{pick}}$ with the feature maps of $\{\hat{I}_{v_{\text{place}}}\}^V_{v_{\text{place}}=1}$ for action-value prediction. We refer readers to [9] for details on this coupling.

### 3.3 Training

We train the action-value function with imitation learning. For each expert demonstration, we construct a tuple $\mathcal{D} = \{\hat{I}_{v^*_{\text{pick}}}, \mathbf{u}^*_{\text{pick}}, \hat{I}_{v^*_{\text{place}}}, \mathbf{u}^*_{\text{place}}\}$, where $\hat{I}_{v^*_{\text{pick}}}$ and $\hat{I}_{v^*_{\text{place}}}$ are the synthesized images whose viewing directions are aligned with the end-effector's rotations; $\mathbf{u}^*_{\text{pick}}$ and $\mathbf{u}^*_{\text{place}}$ are the best pixels in those views annotated by experts. We draw pixels $\{\hat{\mathbf{u}}_j | \hat{\mathbf{u}}_j \neq \mathbf{u}^*_{\text{pick}}, \hat{\mathbf{u}}_j \neq \mathbf{u}^*_{\text{place}}\}^{N_{\text{neg}}}_{j=1}$ from randomly synthesized images $\hat{I}_{\text{neg}}$ as negative samples. For brevity, we omit the subscript for pick and place and present the loss function that is used to train both action-value functions:

$$\mathcal{L}(\mathcal{D}) = -\log p\Big(\mathbf{u}^* | \hat{I}, \hat{I}_{\text{neg}}, \{\hat{\mathbf{u}}_j\}^{N_{\text{neg}}}_{j=1}\Big), \quad p\Big(\mathbf{u}^* | \hat{I}, \hat{I}_{\text{neg}}, \{\hat{\mathbf{u}}_j\}^{N_{\text{neg}}}_{j=1}\Big) = \frac{e^{-E_\theta(\hat{I}, \mathbf{u})}}{e^{-E_\theta(\hat{I}, \mathbf{u})} + \sum^{N_{\text{neg}}}_{j=1} e^{-E_\theta(\hat{I}_{\text{neg}}, \hat{\mathbf{u}}_j)}} \qquad (7)$$

A key innovation of our objective function compared to previous works [9, 10, 36, 67] is the inclusion of negative samples $\{\hat{\mathbf{u}}_j\}^{N_{\text{neg}}}_{j=1}$ from imagined views $\hat{I}_{\text{neg}}$. We study the effects of ablating negative samples in Sec. 3.4 and show that they are essential for successfully training action-value functions. In practice, we batch the image that contains the positive pixel and other synthesized views for the forward pass. Every pixel in these images are treated as samples to compute the loss.

We execute experiments in both simulation and real-world settings to evaluate the proposed method across various tasks.

### 3.4 Simulation Experiments

**Environment.** We propose four new 6-DoF tasks based on Ravens [9] and use them as the benchmark. We show qualitative examples of these tasks in Fig. 4 and summarize their associated challenges in the supplementary materials. Notably, place-red-in-green is a relatively cluttered environment where 5-10 distractor objects are randomly spawned and placed; hanging-disks and stacking-objects require the policy to generalize to novel objects that are not seen during training. All simulated experiments are conducted in PyBullet [68] using a Universal Robot UR5e with a suction gripper. The input observations for MIRA are 30 RGB images from different cameras pointing toward the center. For all the baselines, we additionally supply the corresponding noiseless depth images. Each image has a resolution of $640 \times 480$. The camera has focal length $f = 450$ and camera center $(c_x, c_y) = (320, 240)$. Demonstrations are collected with motion planner that can access the ground-truth states of objects.

**Evaluation.** For each task, we perform evaluations under two settings: *in-distribution* configures objects with random rotations ($\theta_x, \theta_y \in [-\frac{\pi}{6}, \frac{\pi}{6}], \theta_z \in [-\pi, \pi]$). This is also the distribution we used

to construct the training set. *out-of-distribution* instead configures objects with random rotations $(\theta_x, \theta_y \in [-\frac{\pi}{4}, -\frac{\pi}{6}] \cup [\frac{\pi}{6}, \frac{\pi}{4}], \theta_z \in [-\pi, \pi])$. We note that these rotations are outside the training distribution and also correspond to larger out-of-plane rotations. Thus, this setting requires stronger generalization. We use a binary score (0 for failure and 1 for success) and report results on 100 evaluation runs for agents trained with $n = 1, 10, 100$ demonstrations.

**Baseline Methods.** Although our method only requires RGB images, we benchmark against published baselines that additionally require depth images as inputs. Form2Fit [14] predicts the placing action by estimating dense descriptors of the scene for geometric matching. Transporter-*SE*(2) and Transporter-*SE*(3) are both introduced in Zeng et al. [9]. Although Transporter-*SE*(2) is not designed to solve manipulation tasks that require 6-DoF actions, its inclusion helps indicate what level of task success can be achieved on the shown tasks by simply ignoring out-of-plane rotations. Transporter-*SE*(3) predicts 6-DoF actions by first using Transporter-*SE*(2) to estimate *SE*(2) actions, and then feeding them into a regression model to predict the remaining rotational $(r_x, r_y)$ and translational (z-height) degrees of freedom. Additionally, we benchmark against a baseline, GT-State MLP, that assumes perfect object poses. It takes ground truth state (object poses) as inputs and trains an MLP to regress two $SE(3)$ poses for $\mathcal{T}_{\text{pick}}$ and $\mathcal{T}_{\text{place}}$.

| Method | block-insertion in-distribution-poses | | | block-insertion out-of-distribution-poses | | | place-red-in-greens in-distribution-poses | | | place-red-in-greens out-of-distribution-poses | | |
|---|---|---|---|---|---|---|---|---|---|---|---|---|
| | 1 | 10 | 100 | 1 | 10 | 100 | 1 | 10 | 100 | 1 | 10 | 100 |
| GT-State MLP | 0 | 1 | 1 | 0 | 0 | 1 | 0 | 1 | 3 | 0 | 1 | 1 |
| Form2Fit [14] | 0 | 1 | 10 | 0 | 0 | 0 | **35** | 79 | **96** | 21 | 30 | 61 |
| Transporter-*SE*(2) [9] | 25 | 69 | 73 | **1** | 21 | 20 | 30 | 74 | 83 | **25** | 18 | 36 |
| Transporter-*SE*(3) [9] | **26** | 70 | 77 | 0 | 20 | 22 | 29 | 77 | 85 | 23 | 20 | 38 |
| Ours | 0 | **84** | **89** | 0 | **74** | **78** | 27 | **89** | **96** | 22 | **56** | **77** |
| Method | hanging-disks in-distribution-poses | | | hanging-disks out-of-distribution-poses | | | stacking-objects in-distribution-poses | | | stacking-objects out-of-distribution-poses | | |
| | 1 | 10 | 100 | 1 | 10 | 100 | 1 | 10 | 100 | 1 | 10 | 100 |
| GT-State MLP | 0 | 0 | 3 | 0 | 0 | 1 | 0 | 1 | 1 | 0 | 0 | 0 |
| Form2Fit [14] | 0 | 11 | 5 | **4** | 1 | 3 | 1 | 7 | 12 | 0 | 4 | 5 |
| Transporter-*SE*(2) [9] | 6 | 65 | 72 | 3 | 32 | 17 | 0 | **46** | 40 | **1** | **18** | 35 |
| Transporter-*SE*(3) [9] | 6 | 66 | 75 | 0 | 32 | 20 | 0 | 42 | 40 | 0 | 16 | 34 |
| Ours | **13** | **68** | **100** | 0 | **43** | **71** | **13** | 21 | **76** | 0 | 3 | **74** |

**Table 1: Quantitative results.** Task success rate (mean %) vs. # of demonstration episodes (1, 10, 100) used in training. Tasks labeled with in-distribution configure objects with random rotations $(\theta_x, \theta_y \in [-\frac{\pi}{6}, \frac{\pi}{6}], \theta_z \in [-\pi, \pi])$. This is also the rotation distribution used for creating the training set. Tasks labeled with out-of-distribution configure objects with rotations $(\theta_x, \theta_y \in [-\frac{\pi}{4}, -\frac{\pi}{6}] \cup [\frac{\pi}{6}, \frac{\pi}{4}], \theta_z \in [-\pi, \pi])$ that are (i) outside the training pose distribution and (ii) larger out-of-plane rotations.

**Results.** Fig. 5 shows the average scores of all methods trained on different numbers of demonstrations. GT-State MLP fails completely; Form2Fit cannot achieve 40% success rate under any setting; Both Transporter-*SE*(2) and Transporter-*SE*(3) are able to achieve $\sim$70% success rate when the object poses are sampled from the training distribution, but the success rate drops to $\sim$30% when the object poses are outside the training distribution.

MIRA outperforms all baselines by a large margin when there are enough demonstrations of the task. Its success rate is $\sim$90% under the setting of *in-distribution* and $\sim$80% under the setting of *out-of-distribution*. The performance improvement over baselines demonstrates its generalization ability thanks to the action optimization process with novel view synthesis and affordance prediction in the loop.

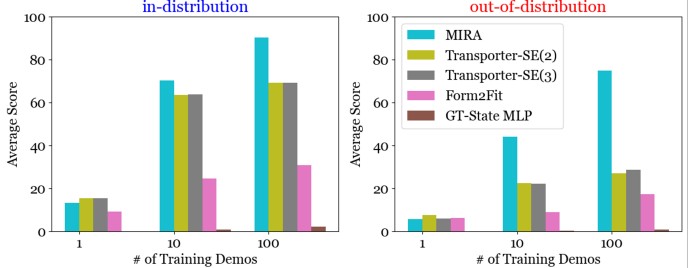

**Figure 5:** Average scores of all methods under both in-distribution and out-of-distribution settings.

Interestingly, we found that MIRA sometimes performs worse than

baselines when only 1 demonstration is supported. We hypothesize that this is because our action-value function needs more data to understand the subtle differences between images rendered from different views in order to select the best view. We show the full quantitative results in Table 1.

**Ablation studies.** To understand the importance of different components within our framework, we benchmark against two variants: (i) ours w/ perspective ray casting, and (ii) ours w/o multi-view negative samples in Eq. (7). We show the quantitative results in Table 2. We find that ours w/ perspective ray casting fail to learn the action-value functions because the perspective images (a) contain distorted appearances of the objects that are challenging for CNNs to comprehend and (b) the groudtruth picking or placing locations may be occluded by the robot arms. We visualize these challenges in the supplementary materials. Our method with orthographic ray casting circumvents both challenges by controlling the near/far planes to ignore occlusions without worrying about the distortion and scaling. Ours w/o multi-view negative samples also fails to learn reliable action-value functions, and this may be due to a distribution shift between training and test: during training, it has only been supervised to choose the best pixel given an image. However, it is tasked to both select the best pixel and best view during the test time.

| | block-insertion in-distribution-poses | | | block-insertion out-of-distribution-poses | | |
|---|---|---|---|---|---|---|
| Method | 1 | 10 | 100 | 1 | 10 | 100 |
| Ours w/ perspective | 0 | 0 | 0 | 0 | 0 | 0 |
| Ours w/o multi-view negatives | 0 | 11 | 13 | 0 | 2 | 4 |
| Ours | 0 | **84** | **89** | 0 | **74** | **78** |

Table 2: **Ablation studies.** We study the effects of ablating orthographic ray casting or multi-view negative samples from our system.

## 3.5 Real-world Experiments

We validate our framework with three kitting tasks in the real world and show qualitative results in Fig. 2. Additional qualitative results and video can be found in the supplementary material. Our system consists of a UR5 arm, a customized suction gripper, and a wrist-mounted camera. We show that our method can successfully (i) pick up floss cases and pack them into transparent containers, (ii) pick up metal cubes and insert them into the cases, and (iii) pick up a stainless steel ice sphere and place it into 10+ different cups configured with random translations and out-of-plane rotations. See Fig. 2 for qualitative results. These tasks are challenging because (i) they include objects with reflective or transparent materials, which makes these tasks not amenable to existing works that require depth sensors [9, 12, 14], and (ii) they require out-of-plane action reasoning. The action-value functions are trained with 20 demonstrations using these cups. Demonstrations are supplied by humans who teleoperate the robot through a customized user interface. At the beginning of each pick-and-place, our system gathers 30 $1280 \times 720$ RGB images of the scene with the wrist-mounted camera. These 30 locations are sampled from a circular path on the table and the robot moves the camera to each of them sequentially through inverse kinematics. Each image's camera pose is derived from the robotic manipulator's end-effector pose and a calibrated transformation between the end-effector and the camera. This data collection procedure is advantageous as industrial robotic manipulators feature sub-millimeter repeatability, which provides accurate camera poses for building NeRF. In practice, we search through $V = 121$ virtual views that uniformly cover the workspace and predict their affordances for optimizing actions. The optimization process currently takes around 2 seconds using a single NVIDIA RTX 2080 Ti GPU. This step can be straightforwardly accelerated by parallelizing the computations with multiple GPUs.

## 4 Limitations And Conclusion

In terms of limitations, our system currently requires training a NeRF of the scene for each step of the manipulation. An instant-NGP [64] requires approximately 10 seconds to converge using a single NVIDIA RTX 2080 Ti GPU, and moving the robot arms around to collect 30 multi-view RGB images of the scene takes nearly 1 minute. This poses challenges to apply MIRA to tasks that require real-time visio-motor control. We believe observing the scene with multiple mounted cameras or learning a prior over instant-NGP could drastically reduce the runtime. In the future, we plan to explore the usage of mental imagery for other robotics applications such as navigation and mobile manipulation.

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
