# OpenReview forum: "MIRA: Mental Imagery for Robotic Affordances"
_robot-learning.org/CoRL/2022/Conference — CoRL 2022 Poster_

### Official Review · Reviewer_jzYJ · 2022-07-27

**Originality:** Excellent
**Technical Quality:** Excellent
**Clarity Of Presentation:** Very Good
**Impact:** 4

**Recommendation:**

Strong Accept: I recommend accepting the paper and will argue for my recommendation even if other reviewers hold a different opinion.

**Summary:**

This paper presents MIRA, a supervised pick-and-place manipulation framework that uses a NeRF-based scene representation to efficiently learn 6-DoF policies. The framework first builds an NeRF of a tabletop scene with 30 images captured from various viewpoints. A number of sampled viewpoints are rendered to generate novel perspectives of the scene. Then, actions are predicted by selecting the best pixel location to pick and place (separately) over all the rendered perspectives. This approach is able to learn robust 6-DoF policies without any depth data, which makes it applicable to objects with reflective, glossy, and translucent surfaces that are typically hard to model with commodity depth sensors. Experimental results include comparisons against TransporterNets, Form2Fit, and ground-truth state MLPs. The results show that MIRA significantly outperforms these baselines, specifically in scenes with out-of-distribution object poses, all without using explicit depth information.

**Issues:**

Mostly additional details regarding the approach. See the Weaknesses section above for details.

**Quality Of The Limitations Section:**

Limitations are addressed clearly

**Reviewer Expertise:**

4: The reviewer is confident but not absolutely certain that the evaluation is correct

**Robotics Focus:**

Sufficient demonstration on hardware

**Strengths And Weaknesses:**

**Strengths**

+ Overall, the idea of using NeRFs with action-centric perception 6-DoF manipulation is quite novel and an interesting direction for the field. Naive image-to-action agents often lack the 3D prior for efficiently learning 6-DoF actions. Even with sufficient data, image-based models struggle to be robust to scene changes. Further, NeRFs also circumvent common issues with depth-cameras like modeling specular, glossy, translucent surfaces, and small objects. While there are some limitations to MIRA's approach, the formulation here has a lot of potential to be extended to general problems in vision-based manipulation. It’s exciting to see how future works build on this approach.

+ The experiments cover a good set of baselines for action-centric perception. These evaluations test settings with both in-distribution and out-of-distribution object poses. The results also include ablations that investigate specific contributions like orthographic projection and multi-view negative sampling for EBM training. The simulated experiments are set in an open-source simulator (Ravens), and should be easily reproducible, especially if the authors release their code and setup. The real-world results are quite compelling, particularly the results with shiny and translucent objects.

+ The writing and presentation of methods is mostly clear. Figure 1 provides a good overview of the framework. Section 3 does a good job of providing preliminaries on NeRFs and orthographic ray-casting. The supplementary video also provides a good overview of the approach and highlights its benefits.

**Weaknesses**

- One key limitation (acknowledged in the limitations section) is that MIRA requires 30 multi-view RGB images, which takes 1 minute to capture with a robot arm. Optimizing the NeRF requires another 10 seconds. While this might be feasible for simple pick-and-place tasks, it’s not easily applicable to dynamic tasks that require real-time visuomotor control. For instance, while opening a cabinet or pouring water into a mug, it’s hard to capture 30 images with the same manipulator that is interacting with the object. However, given the rapid-pace of advancements with neural radiance fields, it might be possible to build 3D scene representations with limited viewpoints like in pixelNeRF (Yu et al. 2020). Nonetheless, the contribution here to quasi-static 6-DoF pick-and-place tasks itself is sufficient and interesting to the robot-learning community.

- Section 3.2 and 3.3 could benefit from additional details on the approach. More specifically, are all the multi-view images batched and processed with the ConvNet in a single-pass? Isn’t this expensive? I also assume it’s one ConvNet with shared weights for all the images, or is each image processed with independent ConvNets and kept spatially consistent across scenes? What is the ratio of positive to negative samples for the EBM training? And how dense are these samples – can you use every pixel in the image as training samples, like in TransporterNets?

- Similarly, Section 4.2 (or an appendix section) could provide additional details on the image-capture process. How are the 30 multi-view images captured for the real-robot experiments? Is this a predefined path executed with IK or a motion-planner? This information might be useful for practitioners to understand how to get a good coverage of the tabletop. Also, were all 121 virtual images crucial for precise 6-DoF actions? Would it be possible to use less virtual images? Perhaps this could be investigated with simulated experiments by varying the number of virtual images and studying how it affects performance in Ravens tasks.

References:

Yu et al. 2020 – https://arxiv.org/abs/2012.02190


**Summary Of Recommendation:**

MIRA presents a novel and interesting approach for using NeRFs with action-centric perception for  pick-and-place manipulation in 6-DoF settings. The ideas and experiments from the paper are very relevant to the CoRL community.

---

> ### Author Response · Authors · 2022-08-20
> **Response to Reviewer jzYJ Part 1**
>
> Thank you for your constructive feedback and important technical questions! The following is our response to your questions. The revised paper can be found in General Response Part 1 with changes highlighted in the cyan color.
>
> Comment: **were all 121 virtual images crucial for precise 6-DoF actions? Would it be possible to use less virtual images?**
> - Response: We have performed additional experiments on using fewer virtual images. The results show that *it is possible to use less virtual images*! MIRA already outperforms all baselines using only 49 virtual images. Here are the full experimental results:
>
>     | Methods                                      | MIRA 9 virtual images | MIRA 25 virtual images | MIRA 49 virtual images | MIRA 81 virtual images | MIRA 121 virtual images | Transporter-SE(3) | Transporter-SE(2) | Form2Fit |
>     | -------------------------------------------- | --------------------- | ---------------------- | ---------------------- | ---------------------- | ----------------------- | ----------------- | ----------------- | -------- |
>     | block-insertion out-of-distribution-poses    | 36                    | 55                     | 71                     | 73                     | 78                      | 22                | 20                | 0        |
>     | place-red-in-green out-of-distribution-poses | 34                    | 49                     | 64                     | 65                     | 77                      | 38                | 36                | 61       |
>     | stacking-objects out-of-distribution-poses      | 46                    | 56                     | 61                     | 71                     | 71                      | 20                | 17                | 3        |
>     | hanging-disks out-of-distribution-poses   | 30                    | 49                     | 66                     | 70                     | 74                      | 34                | 35                | 5        |
> - Update: The rebuttal will be included in the final revision.
>
> Comment: **I also assume it’s one ConvNet with shared weights for all the images, or is each image processed with independent ConvNets and kept spatially consistent across scenes?**
> - Response: We use one ConvNet with shared weights for all the images. This encourages MIRA to be 3D rotational equivariant as synthesized views can densely cover novel orientations of objects. It also helps us verify that MIRA outperforms TransporterNets due to our algorithmic design instead of having a larger model capacity.
> - Update: We have updated Section 3.2 to reflect this.
>
> Comment: **What is the ratio of positive to negative samples for the EBM training?**
> - Response: The positive to negative ratio is 1:102399 in our current implementation. For each training iteration, we sample 2 images: the first image $I_1$ contains the positive pixel (derived from the demonstration), and the other image $I_2$ consists of all negative pixels. We batch these two images and send them into our policy to predict dense affordances. Since each rendered virtual image has a size of 160 x 320, the resulting ratio is 1 : 2 * 160 * 320 - 1.
> - Update: We have updated Section 3.3 to discuss this.
>
> Comment: **how dense are these samples – can you use every pixel in the image as training samples, like in TransporterNets?**
> - Response: Absolutely! We are using every pixel in the image as training samples in order to save computation time.
> - Update: We have updated Section 3.3 to discuss this.
>
> Comment: **How are the 30 multi-view images captured for the real-robot experiments?**
> - Response: We uniformly sample 30 3D locations on a circular path above the table as the camera translations. Then, we determine their view directions by pointing toward the center of the table. For each capture, we command the robot to move to these SE(3) poses sequentially using inverse kinematics.
> - Update: We have updated Section 4.2 to include this.
>
> Comment: **are all the multi-view images batched and processed with the ConvNet in a single-pass?**
> - Response: Yes, we batch the multi-view images (batch size = 4) and run inference on each batch of the virtual images (121 images) sequentially. We believe this step can be drastically accelerated by parallelizing the computation with a multi-GPU setup.
> - Update: We have updated Section 3.2 to discuss this.

---

> > ### Author Response · Authors · 2022-08-20
> > **Response to Reviewer jzYJ Part 2**
> >
> > Comment: **One key limitation (acknowledged in the limitations section) is that MIRA requires 30 multi-view RGB images**
> >
> > - Response: We present new experiments that analyze the number of initial images. Generally, we found that MIRA outperform all baselines given 15 views. For each task, we test MIRA trained with 100 demos and provide it with N $\in$ \{10, 15, 20, 25, 30\} images of the scene:
> >
> >     | Methods                                      | MIRA 10 input images | MIRA 15 input images | MIRA 20 input images | MIRA 25 input images | MIRA 30 input images | Transporter-SE(3) | Transporter-SE(2) | Form2Fit |
> >     | -------------------------------------------- | -------------- | -------------- | -------------- | -------------- | -------------- | ----------------- | ----------------- | -------- |
> >     | block-insertion out-of-distribution-poses    | 3              | 56             | 74             | 77             | 78             | 22                | 20                | 0        |
> >     | place-red-in-green out-of-distribution-poses | 24             | 46             | 71             | 74             | 77             | 38                | 36                | 61       |
> >     | hanging-disks out-of-distribution-poses      | 0              | 31             | 65             | 66             | 71             | 20                | 17                | 3        |
> >     | stacking-objects out-of-distribution-poses                                             | 0               | 49               | 55               |  74              | 74               |  34                 | 35                  | 5         |
> > - Update: The new experiment will be added to the final revision.
> >
> > Comments: **The simulated experiments are set in an open-source simulator (Ravens), and should be easily reproducible, especially if the authors release their code and setup**.
> > - Response: We promise to release our code to the public during the camera-ready phase.
> >
> > Thank you again for your time in reviewing our paper. We hope these responses are reasonable. We also want to thank you for giving us high ratings for all of {Originality, Technical Quality, Clarity of Presentation, Impact}: {Excellent, Excellent, Very Good, and 4/4} which we sincerely appreciate!

---

> > ### Comment · Reviewer_jzYJ · 2022-08-25
> > **Looks good!**
> >
> > The new results with virtual images look good. The authors have addressed all my questions. Happy to keep my score.
> >
> > _Side note on novelty_: The other two reviewers seem to be concerned about MIRA's novelty. I don't think the approach here is just a  straightforward usage of NeRFs. MIRA cleverly integrates a neural renderer into the action-space of a 6-DoF manipulator. This is something I haven't seen before in manipulation (as far as I am aware). Also, just because an approach _uses NeRFs_, doesn't make it less novel, just like there are 1000s of papers that _use_ ResNets or Transformers.

---

> > > ### Author Response · Authors · 2022-08-28
> > > **Thank you!**
> > >
> > > We are happy to know that our response has addressed your questions! Your reviews drastically improved our a) presentation of technical details and b) ablation studies (i.e., how many views are needed). Thanks again.

---

### Official Review · Reviewer_YteW · 2022-07-31

**Originality:** Fair
**Technical Quality:** Good
**Clarity Of Presentation:** Good
**Impact:** 3

**Recommendation:**

Weak Accept: I recommend accepting the paper, but will not argue for my recommendation if the majority of other reviewers have a different opinion.

**Summary:**

This paper uses a state-of-the-art neural representation, i.e., neural radiance fields (nerfs), to plan robot manipulation actions. Using a set of 2D images, acquired from a diverse range of angles/poses, a 3D volume of the scene is optimized, from which an orthographic view is rendered and fed into a policy that predicts a pick-and-place action value. In essence, the diverse set of initially acquired views allows synthesizing new views that might result in selecting a better manipulation position.


**Issues:**

- an in-depth analysis on the effect of number of initial images.
- related work: Nerf for Robotics subsection needs to provide a context as to the difference of the proposed approach to the mentioned prior work.
- related work: how does this work compare to [R1,2]
- where do expert demonstrations come from?
- evaluation of and discussion on how/whether the same approach is applicable to 2/3-finger based grasping/picking actions.

[R1] D. Driess et al., "Learning Multi-Object Dynamics with Compositional Neural Radiance Fields", arXiv preprint, 2022.
[R2] D. Driess et al., "Reinforcement Learning with Neural Radiance Fields", arXiv preprint, 2022.

**Quality Of The Limitations Section:**

Limitations are addressed clearly

**Reviewer Expertise:**

4: The reviewer is confident but not absolutely certain that the evaluation is correct

**Robotics Focus:**

Sufficient demonstration on hardware

**Strengths And Weaknesses:**

strengths:
- a good use-case of nerfs for robotic manipulation
- a good range of evaluations, including real robot experiments

weaknesses:
- no methodological novelty
- too many views initially required to build nerf
- too long optimization/computation time prior to action prediction + very long time to gather the initial views -> infeasible for realistic manipulation tasks


**Summary Of Recommendation:**

The paper investigates a straight-forward application of neural representations for robot manipulation. In particular, neural radiance fields (nerfs) are optimized from multiple views that allows constructing the 3D volume of the scene, which in turn enable better manipulation action planning. The evaluations and the experiments cover a good range and provides convincing evidence on the usability of the approach. The major drawbacks stand as the computation time and the initial time cost of acquiring 30 images from different views, both of which diminishes the value proposition of the work significantly.

---

> ### Author Response · Authors · 2022-08-19
> **Response to Reviewer YteW Part 1**
>
> Thank you for the helpful and insightful feedback! The following is our response to your helpful comments. The revised paper can be found in General Response Part 1 with changes highlighted in the cyan color.
>
> Comment: **an in-depth analysis on the effect of number of initial images.**
> - Response: We show new experiments that analyze the number of initial images. Generally, we found that MIRA outperform all baselines given 15 views. For each task, we test MIRA trained with 100 demos and provide it with N $\in$ \{10, 15, 20, 25, 30\} images of the scene:
>
>     | Methods                                      | MIRA 10 images | MIRA 15 images | MIRA 20 images | MIRA 25 images | MIRA 30 images | Transporter-SE(3) | Transporter-SE(2) | Form2Fit |
>     | -------------------------------------------- | -------------- | -------------- | -------------- | -------------- | -------------- | ----------------- | ----------------- | -------- |
>     | block-insertion out-of-distribution-poses    | 3              | 56             | 74             | 77             | 78             | 22                | 20                | 0        |
>     | place-red-in-green out-of-distribution-poses | 24             | 46             | 71             | 74             | 77             | 38                | 36                | 61       |
>     | hanging-disks out-of-distribution-poses      | 0              | 31             | 65             | 66             | 71             | 20                | 17                | 3        |
>     | stacking-objects out-of-distribution-poses          | 0               | 49               | 55               |  74              | 74               |  34                 | 35                  | 5         |
> - Update: The new experiments will be included in the final revision.
>
> Comment: **where do expert demonstrations come from?**
> - Response: For real-world experiments, we asked humans to perform teleoperation using a customized user interface. For simulation experiments, we accessed the ground truth states of objects and plan the action with a motion planner.
> - Update: We have updated both Sections 4.1 and 4.2 of the paper to clarify this.
>
> Comment: **evaluation of and discussion on how/whether the same approach is applicable to 2/3-finger based grasping/picking actions.**
> - Response: The same approach can be extended to any end-effector as long as its state is specified with 6-DoF poses. Our formulation does not make any assumption that is specific to suction grippers.
>
> Comment: **related work: Nerf for Robotics subsection needs to provide a context as to the difference of the proposed approach to the mentioned prior work.**
> - Response: The revised paper is attached in General Response Part 1. The following is the updated paragraph:
>     > Neural fields have emerged as a promising tool to represent 2D images [40], 3D geometry [41, 42], appearance [6, 43, 44], touch [45], and audio [46, 47]. They offer several advantages over classic representations (e.g., voxels, point clouds, and meshes) including reconstruction quality, and memory efficiency. Several works have explored the usage of neural fields for robotic applications including localization [48, 49], SLAM [50, 51, 52], navigation [53], dynamics modeling [54, 55, 56, 57], and reinforcement learning [58]. For robotic manipulation, GIGA [59] jointly trains a grasping network and an occupancy network for synergy; NDF [12] uses the features of occupancy networks [42] as object descriptors for few-shot imitation learning. Both GIGA [59] and NDF [12] rely on depth cameras to perceive the 3D geometry of the scene, while MIRA does not require depth sensors and thus can handle objects with reflective or thin-structured materials. Dex-NeRF [60] infers the geometry of transparent objects with NeRF and determines the grasp poses with Dex-Net [30]. However, it only predicts the 3-DoF grasping pose and does not provide a solution for pick-conditioned placing; NeRF-Supervision [61] uses NeRF as a dataset generator to learn dense object descriptors for picking but not placing. In contrast, MIRA is capable of predicting both 6-DoF picking and pick-conditioned placing. Model-based methods [54,55] use NeRFs as decoders to learn latent state representations for model predictive control; NeRF-RL [58] instead uses the learned latent state representations for downstream reinforcement learning. Compared to these works, MIRA uses imitation learning to acquire the policy and thus avoid the conundrum of constructing reward functions. Furthermore, MIRA enjoys better sample efficiency due to the approximate 3D rotational equivariance via novel-view synthesis (Section 3.2).
> - Update: We have updated our subsection 2.2 Neural Fields for Robotics to incorporate this.

---

> > ### Author Response · Authors · 2022-08-20
> > **Response to Reviewer YteW Part 2**
> >
> > Comment: **related work: how does this work compare to [R1,2]**
> > - Response: [R1] is a model-based method that uses NeRFs as decoders to learn latent state representations for model predictive control; [R2] instead uses the learned latent state representations for downstream reinforcement learning. Compared to these works, MIRA uses imitation learning to acquire the policy and thus avoid the conundrum of constructing reward functions. Furthermore, MIRA enjoys better sample efficiency due to the approximate 3D rotational equivariance via novel-view synthesis (Section 3.2).
> > - Update: We have updated our subsection 2.2 Neural Fields for Robotics to reflect this.
> >
> > Comment: **too long optimization/computation time prior to action prediction**
> > - Response: We present additional experiments on using fewer virtual images for shorter optimization/computation time. The fewer virtual images MIRA uses, the shorter computation/optimization time because a) instant-NeRF needs to render fewer images and b) the policy has fewer views to consider. Our results show that MIRA already outperforms all baselines using only 49 virtual images. Here are the full experimental results:
> >
> >     | Methods                                      | MIRA 9 virtual images | MIRA 25 virtual images | MIRA 49 virtual images | MIRA 81 virtual images | MIRA 121 virtual images | Transporter-SE(3) | Transporter-SE(2) | Form2Fit |
> >     | -------------------------------------------- | --------------------- | ---------------------- | ---------------------- | ---------------------- | ----------------------- | ----------------- | ----------------- | -------- |
> >     | block-insertion out-of-distribution-poses    | 36                    | 55                     | 71                     | 73                     | 78                      | 22                | 20                | 0        |
> >     | place-red-in-green out-of-distribution-poses | 34                    | 49                     | 64                     | 65                     | 77                      | 38                | 36                | 61       |
> >     | stacking-objects out-of-distribution-poses      | 46                    | 56                     | 61                     | 71                     | 71                      | 20                | 17                | 3        |
> >     | hanging-disks out-of-distribution-poses   | 30                    | 49                     | 66                     | 70                     | 74                      | 34                | 35                | 5        |
> > - Update: The new experiments will be added to our paper in the final revision.
> >
> >
> > We would love to thank you again for reviewing our paper. The experiments you suggested (e.g., analyzing the number of input views) have helped us make the paper more complete. We hope these responses have addressed your questions!

---

### Official Review · Reviewer_DpD8 · 2022-08-05

**Originality:** Good
**Technical Quality:** Very Good
**Clarity Of Presentation:** Excellent
**Impact:** 3

**Recommendation:**

Weak Reject: I recommend rejecting the paper, but will not argue for my recommendation if the majority of other reviewers have a different opinion.

**Summary:**

The method enables the robot to apply 6-DOF pick-up and place actions given RGB-only image of the environment with objects. Instead of directly learning 6 DOF pose, the system learns to predict the best-projected plane of the scene that enables an orthogonal approach of the end-effector to this projected plane for successful pick-up and place. NeRFs enable rendering of the scene with orthogonal projection from any arbitrary camera pose, given a number of scenes from different camera poses. NeRFs is used to generate a number of projections from sampled camera poses, and the problem is reduced to learning affordances and 3 DOF (x,y,yaw) parameters of pick-up and place actions.

The method is verified in the simulator in different environments, and in the real robot in an insertion task. The performance is shown to surpass the performance of a number of SOTA methods in the simulator. The method does not require depth information, yet achieves good performance in 6 DOF action execution.

**Issues:**

- Is is possible to compare with other NeRF based control methods? [54] makes 6 DOF control. Can [60] be extended to 6DOF control?
- Is taking 30 snapshots from different camera poses feasible and scalable in the real world?
- How well does the method perform in cluttered environments?

**Quality Of The Limitations Section:**

Additional details required

**Reviewer Expertise:**

3: The reviewer is fairly confident that the evaluation is correct

**Robotics Focus:**

Sufficient demonstration on hardware

**Strengths And Weaknesses:**

The strenghts of the paper:
- Instead of RGB-D data, the method works with RGB only data. This enables the system to work with reflective and this structured objects.
- Object detection is not required. The affordances are learned in pixel level.
- Orthogonal projection enables better affordance prediction compared to perspective projection.
- The method works well in out-of-distribution validation cases compared to other methods.
- The paper is very well written.
- The method is verified with experiments that require generalization capabilities.

The weak points in the paper:
- Many stated contributions can be attributed to the existing NeRF method. As such, the used scene representation created by NeRF is not novel. The orthogonal projection is not challenging with NeRFs. Assigning affordances to pixels of scenes is also not novel.
- The paper combines the following: Generating a number of projected scenes using NeRF and assigning affordances to the pixels in these scenes. This approach is smart, but it corresponds to an engineering effort of combining existing SOTA methods, rather than proposing a highly original/novel method with significant impact to the field.
- The method is compared with baselines such MLP, Form2Fit, Transporter-SE(2/3). This paper out-performed the other methods probably because of the underlying NeRFs method, however NeRF is not a contribution of this paper. In order to better identify/show the contribution, the paper could have been compared with other existing control methods that use NeRF such as [54] or [60].
- There is no object detection step, yet, the method was not verified in cluttered settings with more complex actions (such as grasping with a gripper).


**Summary Of Recommendation:**

The paper utilizes the NeRF in a smart way in order to create potential projections of an arbitrary environment and reduces 6DOF end-effector pick-up/place pose search into a search of projections + orthogonal approach with 3DOF for pick-up/place. Although this is clever and shown to be effective compared to the existing methods, this corresponds to a combination of existing methods rather than a proposal of an original method with high impact, theoretically. Practically, as the contribution is on the control part, the proposed system needed to be compared with other NeRF-based methods.

---

> ### Author Response · Authors · 2022-08-20
> **Response to Reviewer DpD8 Part 1**
>
> **Comment:**
>
> Thank you for your detailed comments and feedback. We provide clarifications, answers, and how we update the paper to each of your concerns below. The revised paper is attached in General Response Part 1. Please let us know if you have any additional questions -- we are happy to clarify or provide additional experiments.
>
> Comment: **Is taking 30 snapshots from different camera poses feasible and scalable in the real world?**
> - Response:
>      1. We believe taking 30 snapshots in the real world is computationally feasible as it takes roughly a minute on a real robot (Section 5).
>      2. We provide additional analysis to see whether MIRA can use fewer views. Our results show that 15 input views are sufficient for MIRA to work reasonably well, which further reduces the data collection time by half. Here are the experimental results:
>
>         | Methods                                      | MIRA 10 images | MIRA 15 images | MIRA 20 images | MIRA 25 images | MIRA 30 images | Transporter-SE(3) | Transporter-SE(2) | Form2Fit |
>         | -------------------------------------------- | -------------- | -------------- | -------------- | -------------- | -------------- | ----------------- | ----------------- | -------- |
>         | block-insertion out-of-distribution-poses       | 3  | 56 | 74  | 77  | 78  | 22 | 20    | 0  |
>         | place-red-in-green out-of-distribution-poses | 24  | 46   | 71 | 74 | 77 | 38  | 36  | 61 |
>         | hanging-disks out-of-distribution-poses        | 0   | 31  | 65 | 66 | 71  | 20 | 17  | 3  |
>         | stacking-objects out-of-distribution-poses    | 0  | 49   | 55  |  74 | 74  |  34  | 35 | 5 |
>
>    3. Furthermore, there are orthogonal works that tackle the problem of building NeRFs with fewer input views. For example, RegNeRF is able to generate high-quality NeRFs with 3 views and pixelNeRF shows results in building NeRFs with 1 view.
>       - RegNeRF, Niemeyer et al., https://arxiv.org/abs/2112.00724
>       - pixelNeRF, Yu et al., https://arxiv.org/abs/2012.02190
>   - Update: The new experiments will be added in the final revision.
>
> Comment: **The method is verified in the simulator in different environments, and in the real robot in an insertion task**.
> - Response: We would like to clarify that our approach has actually been shown on the real robot on 3 separate tasks:
>   1. Place the floss into its container.
>   2. Place the metal ball into different cups.
>   3. Place the metal cubes into their container.
>
>   The video and task details can be found in the supplementary material.
>
> - Update: We have revised Section 4.2 of the paper to reflect this.
>
> Comment: **How well does the method perform in cluttered environments?**
>
> - Response: We have tested our method on cluttered environments using a simulation task, *place-red-in-green*, that is relatively cluttered compared to environments presented in the previous work [54]. In this task, 5 to 10 distractor objects are randomly spawned and placed inside the scene. These distractor objects have random shapes and colors, and they may occlude the relevant objects (the red block and green bowls) from certain views. The reader can watch the test videos on cluttered scenes by downloading the attached Zip file below. It contains 5 recorded MIRA test instances for the task *place-red-in-green*.
>
> - Update: We have revised Section 4.1 of the paper to discuss this.
>
> Comment: **The orthogonal projection is not challenging with NeRFs**.
> - Response: To our knowledge, all existing work on NeRFs train and render images using a perspective camera model. In contrast, our work is the first to demonstrate that when a NeRF model is trained on perspective cameras, it may directly be utilized to render orthographic images. Such a result is both non-obvious and surprising because rays are now rendered using a *different volume rendering procedure* from the one used during training time, and the orthographic rendering process essentially corresponds to an out-of-distribution rendering test on the NeRF model. Simultaneously, being able to render virtual cameras using orthographic projection is useful for manipulation, as it enables us to ignore occlusions of the robotic scene (and is impossible to physically construct in the real world using a single camera).
>   Thus, we believe that our approach toward orthographic rendering is a novel, non-trivial, and useful finding that is worth sharing with the community.
>
> - Update: We have updated Section 3.1 to include the above response.
>
>
> **Zip File:**
>
> /attachment/cce2e006b2a2491c8cd2daaa844854e2c0e8a4d9.zip

---

> > ### Author Response · Authors · 2022-08-20
> > **Response to Reviewer DpD8 Part 2**
> >
> > Comment: **Although this is clever and shown to be effective compared to the existing methods, this corresponds to a combination of existing methods rather than a proposal of an original method with high impact, theoretically**.
> > - Response: We believe that the proposed method of MIRA contains two new ideas that may be more broadly applied to future papers.
> >   1. We demonstrate and illustrate the idea that optimization can be utilized as a flexible tool to enable generalizable few-shot manipulation. In contrast to existing work which enables 6-DoF manipulation using explicit architectural biases (e.g., TransporterNets-SE(3)) or object-centric coordinates systems (e.g., NDF), our approach, through optimization over virtual views, enables generic 6-DoF manipulation on unsegmented scenes with complex motions. These ideas of optimization may further be extended in future work to generalizable manipulation across additional manipulation tasks.
> >   2. We propose the idea that NeRFs may be utilized as a generic tool on which we may render *virtual views* of a scene, even if the underlying camera rendering procedure is different from the one images of the scene are gathered with. This idea can be more broadly extended to render different virtual views of scenes, such as overhead views of rooms and houses to enable more effective navigation in robotics, when the room is captured only by several cameras at the corners of the rooms in the house.
> >
> > Comment: **Is it possible to compare with other NeRF based control methods [54, 60]?**
> > - Response: A direct comparison with [54] is not possible as the authors have not released their implementation to the public. In the following, we highlight several key differences and our major advantages over [54]:
> >     1. **[54] does not show 6-DoF control results**. The simulation experiments presented in [54] are 2-DoF (FluidShake) and 4-DoF (FluidPour). It is not clear whether the method presented in [54] can actually scale to 6-DoF. All the simulation and real-world tasks presented in our work require 6-DoF control.
> >     2. **[54] does not show real-world control results**. In our work, we show results on 3 real-world tasks with shiny and translucent objects (as demonstrated in the supplementary video).
> >     3. **[54] does not show results on generalizing to novel objects**. In contrast, our simulation results on both *hanging-disk* and *stacking-objects* show that our method can generalize to novel objects not seen during training.
> >     4. **[54] requires 2000 times more data to train.** For both FluidShake and FluidPour presented in [54], it uses 1000 * 300 * 20 images (1000 demos, each demo with 300 frames from 20 cameras) to train the model. On the contrary, our model learns reasonable policies with 10 * 30 images (10 demos, each demo contains 1 frame from 30 cameras).
> >     5. **[54] does not show results when the task contains distractor objects that make the task more challenging**. Both FluidShake and FluidPour presented in [54] focus on relatively simple setups where the scene only contains 1 or 2 task-relevant objects. In our simulation task *place-red-in-green*, we show that MIRA can handle tasks that contain multiple distractor objects in the scene. See Figure 3b for an example.
> >     6. **[54] assumes objects' initial poses are configured similarly during the training and test phase**. Therefore, the learned models in [54] do not show generalization results in scenes where objects are configured with very different object poses. In our work, we show that the learned policy can generalize to test scenes where objects have drastically different translations and rotations compared to the trained scenes.
> >
> >     ---
> >
> >     In the following, we discuss why it is not possible/necessary to compare against [60]:
> >     1. **The control part of [60] does not support 6-DoF picking and pick-conditioned placing**. Since the control part of [60], i.e., Dex-Net, focuses on ranking *planar grasp candidates* with a CNN, it is not clear how to extend it to perform 6-DoF picking and pick-conditioned placing.
> >     2. **All baselines in our work receive ground-truth depths. These are the best possible predictions of [60]'s perception part**. We note that [60] proposed to estimate depths with NeRFs and use them as Dex-Net's inputs. Since we directly feed ground-truth depths to all baselines in our simulation benchmark, we are essentially comparing against pick-and-place systems that take the best possible predictions of [60]'s perception part as inputs.
> >     3. **[60] takes longer than 1 hour to train NeRFs and estimate depths**. In MIRA, we adopt instant-NeRF that can train NeRFs in 10 seconds.
> > - Update: We have updated Section 2.2 to reflect this.
> >
> >
> > We want to thank you again for your time in reviewing our paper. The questions you asked have helped us improve the paper significantly. We hope these responses have addressed your concerns!

---

### Meta-Review · Area_Chair_AG7o · 2022-08-15

**Recommendation:** Accept (Poster)
**Confidence:** 4

**Metareview:**

The paper presents a supervised pick-and-place manipulation framework that uses a NeRF-based scene representation to efficiently learn 6-DoF policies. The paper is well written and guides the reader well. It has a good technical quality. The paper is more towards solving an important application using multitude of appropriately combined tools.  The reviewer concerns were adequately addressed. With the current experiments, the paper makes sufficient contributions to the community. .


**Best Paper Nomination:**

No

---

> ### Author Response · Authors · 2022-08-21
> **General Response Part 1**
>
> We sincerely thank all reviewers for their constructive feedback which has helped us improve the paper. **The revised paper (with attached PDF in this comment) incorporates the suggestions with changes highlighted in cyan color**.
>
> We are very excited that reviewers find our work contains convincing results in both simulation and real-world tasks (“... the experiments cover a good range and provides convincing evidence on the usability of the approach” [YteW], "The real-world results are quite compelling, particularly the results with shiny and translucent objects." [jzYJ]), well written ("The paper is very well written." [DpD8]), and novel ("MIRA presents a novel and interesting approach for using NeRFs with action-centric perception ..." [jzYJ]).
>
> ## 1. General clarifications
>
> ### 1.1 Computation time
> The reviewers have expressed concerns about our computation time and suggest that additional experiments and analyses on this subject will make the paper stronger and clarify potential limitations. We have performed the following experiments to improve our paper with new quantitative results.
> 1. **Varying the number of initial input views**. We conduct this experiment to better understand the input data requirement and the results show that MIRA outperforms all baselines given 15 input views. These results show that the data collection time can be reduced by 50%.
>
>     | Methods                                      | MIRA 10 input images | MIRA 15 input images | MIRA 20 input images | MIRA 25 input images | MIRA 30 input images | Transporter-SE(3) | Transporter-SE(2) | Form2Fit |
>     | -------------------------------------------- | -------------- | -------------- | -------------- | -------------- | -------------- | ----------------- | ----------------- | -------- |
>     | block-insertion out-of-distribution-poses | 3 | 56 | 74  | 77  | 78  | 22 | 20 | 0 |
>     | place-red-in-green out-of-distribution-poses | 24  | 46   | 71 | 74 | 77 | 38  | 36  | 61 |
>     | hanging-disks out-of-distribution-poses | 0   | 31  | 65 | 66 | 71  | 20 | 17  | 3  |
>     | stacking-objects out-of-distribution-poses    | 0  | 49   | 55  |  74 | 74 |  34  | 35 | 5 |
>
> 2. **Varying the number of virtual views**. We conduct this experiment to characterize the computational requirement of the policy's inference procedure. Our results show that MIRA retains the performance using 49 virtual images, reducing the original policy inference time by 2.5 times.
>
>     | Methods | MIRA 9 virtual images | MIRA 25 virtual images | MIRA 49 virtual images | MIRA 81 virtual images | MIRA 121 virtual images | Transporter-SE(3) | Transporter-SE(2) | Form2Fit |
>     | -------------------------------------------- | --------------------- | ---------------------- | ---------------------- | ---------------------- | ----------------------- | ----------------- | ----------------- | -------- |
>     | block-insertion out-of-distribution-poses    | 36                    | 55 | 71 | 73  | 78 | 22 | 20 | 0        |
>     | place-red-in-green out-of-distribution-poses | 34                    | 49 | 64 | 65 | 77 | 38 | 36 | 61 |
>     | stacking-objects out-of-distribution-poses | 46 | 56 | 61                     | 71 | 71 | 20 | 17 | 3 |
>     | hanging-disks out-of-distribution-poses   | 30                    | 49 | 66 | 70 | 74 | 34 | 35 | 5 |
>
> ### 1.2 Novelty
> We believe the proposed method of using NeRFs for the action-centric perception of 6-DoF manipulation is interesting for the community and not just a trivial combination of existing methods. While we indeed exploit NeRF and affordances in MIRA (i.e., leverage NeRFs as 3D scene representations, adopt CNN for affordance prediction, etc), why they are used and how they are used are all carefully designed and extensively evaluated. We also propose novel ideas that have not been presented before such as a) framing the 6-DoF manipulation problem as an optimization procedure that searches over synthesized views and their affordances and b) rendering NeRF with orthographic ray casting although it's trained with perspective images. The resulting framework is generic, interpretable, and has been shown to work well in a good range of tasks including real-world experiments. We believe exploiting existing algorithms to realize a novel framework (i.e., MIRA) and results (i.e., 6-DoF pick-and-place) does not mean it is not original or has no technical contribution.

---

> > ### Author Response · Authors · 2022-08-21
> > **General Response Part 2**
> >
> > ## 2. Writing
> > We have revised the paper following the reviewers' suggestions (attached in General Response Part 1, changes highlighted in cyan color). We thank all reviewers for suggestions regarding our writing and clarity. In our revised paper, we made the following changes:
> >
> > 1. We have added clarifications on our simulation and real-world experiments in Section 4 [DpD8].
> > 2. We have added additional discussion on applying our method to dynamic scenes that require real-time visio-motor control in the limitation section [jzYJ].
> > 3. For Related Works, we have provided an additional discussion on how our work compares with previous works including [R1,2] mentioned by [DpD8, YteW].
> > 4. We have provided additional details about our model in Sections 3.2 and 3.3 and the image capturing process in Section 4.2 [jzYJ].
> >
> > We believe the suggested clarifications will improve the communication of our work. The additional experiments will also be included in the final version as well.
> >
> > ## 3. Conclusion
> > We thank the reviewers for their careful feedback and additional suggestions for evaluation, which will make the paper significantly stronger.